# Outcomes at 10-Year Follow-Up after Roux-en-Y Gastric Bypass, Biliopancreatic Diversion, and Sleeve Gastrectomy

**DOI:** 10.3390/jcm12154973

**Published:** 2023-07-28

**Authors:** Georgios-Ioannis Verras, Francesk Mulita, Sjaak Pouwels, Chetan Parmar, Nikolas Drakos, Konstantinos Bouchagier, Charalampos Kaplanis, George Skroubis

**Affiliations:** 1Department of Surgery, General University Hospital of Patras, 26504 Patras, Greece; nikolasdrakos@hotmail.com (N.D.); kbouchagier@windowslive.com (K.B.); xariskaplanis11@hotmail.gr (C.K.) skroubis@med.upatras.gr (G.S.); 2Department of General, Abdominal and Minimally Invasive Surgery, Helios Klinikum, 47805 Krefeld, Germany; sjaakpwls@gmail.com; 3Department of Intensive Care Medicine, Elisabeth-Tweesteden Hospital, 5022 Tiburg, The Netherlands; 4The Wittington Hospital NHS Trust, London N19 5NF, UK; drcparmar@gmail.com

**Keywords:** morbid obesity, obesity surgery, gastric bypass, sleeve gastrectomy, Roux-en-Y bypass

## Abstract

Introduction: Morbid obesity is a well-defined chronic disease, the incidence of which is constantly rising. Surgical treatment of morbid obesity has produced superior outcomes compared to conventional weight loss measures. Currently, there is a gap in the literature regarding long-term outcomes. Our single-institution, retrospective cohort study aims to evaluate weight loss outcomes, comorbidity reduction, and adverse effects at 10 years following Roux-en-Y Gastric Bypass (RYGB), Biliopancreatic Diversion (BPD), and Sleeve Gastrectomy (SG). Materials and Methods: We included all consecutive patients with 10-year follow-up records operated on within our institution. The comparison was carried out on the average percentage of weight and BMI loss. Nausea and vomiting were evaluated through self-reporting Likert scales. Diabetes resolution and nutritional deficiencies were also evaluated. Results: A total of 490 patients from 1995 up to 2011 were included in our study. Of these, 322 underwent RYGB, 58 underwent long-limb BPD, 34 underwent laparoscopic RYGB with fundus excision, 47 underwent laparoscopic SG, and 29 underwent laparoscopic RYGB as a revision of prior SG. RYGB and BPD were significantly associated with higher percentages of weight loss (37.6% and 37.5%), but were not found to be independent predictors of weight loss. Nausea and vomiting were associated with SG and laparoscopic RYGB with fundus excision, more so than the other operations. No differences were observed regarding diabetes resolution and nutritional deficiencies. Conclusions: Longer follow-up reports are important for the comparison of outcomes between different types of bariatric operations. BPD and RYGB resulted in superior weight loss, with no observed differences in diabetes resolution and adverse outcomes.

## 1. Introduction

According to the World Health Organization, obesity is characterized as a chronic illness pertaining to adults, adolescents, and children worldwide, defined by a measured Body Mass Index (BMI) ≥ 30 kg/m^2^. It is estimated that 650 million adults are classified as obese worldwide, representing a staggering 13% of the adult population [1]. Despite obesity being a well-known ailment with a recognized impact on morbidity and mortality, the direction of treatment has largely stayed the same and is centered on weight loss. While most people with obesity will achieve weight reduction through lifestyle and dietary modifications, a percentage of those classified as morbidly obese (BMI ≥ 40 kg/m^2^) will eventually require surgical intervention to minimize their mortality risk. In 2017, approximately 228,000 bariatric operations were performed in the U.S., with the estimated prevalence constantly on the rise [1,2,3].

The long-term monitoring of bariatric patients is crucial in order to evaluate weight loss outcomes, as well as look into a number of obesity-surgery-related adverse effects and the improvement of obesity-related comorbidities such as Gastro-Esophageal Reflux Disease (GERD) and diabetes. While there have been several published randomized controlled trials (RCTs) regarding weight-loss outcomes in paired bariatric surgical approaches, there is currently a gap in the literature regarding long-term follow-up data that compare outcomes and adverse effects for several different surgical operations. The aim of our study is to compare the effectiveness of bariatric surgery (RYGB, BPD, and SG), as well as evaluate its safety and efficacy after 10 years of follow-up in a single institution.

## 2. Materials and Methods

A retrospective search of our unit’s patient database was carried out to identify all consecutive patients that underwent any type of bariatric surgery and had complete 10-year follow-up data. Bariatric surgery was started in our unit in 1995, and hence, data were collected up to 2011 to fulfill the 10-year follow-up data criteria. We included patients that adhered to postoperative follow-up for data including their demographics and weight loss metrics, such as percentage of weight loss and pre- and post-operative BMI. In addition, we included recorded data on a number of obesity-surgery-related adverse effects. The Institutional Review Board granted an exemption of ethics committee approval due to local regulations, as the participants underwent clinical and clinimetric examinations according to routine protocols used at the recruiting hospital and no further procedures were needed.

Our retrospective patient cohort comprised five patient subgroups, depending on the type of bariatric operation. We compared the results of patients undergoing: (1) Laparoscopic Roux-en-Y Gastric Bypass (RYGB); (2) Long-limb Biliopancreatic Diversion (LL-BPD); (3) Laparoscopic RYGB with gastric fundus excision; (4) Laparoscopic Sleeve Gastrectomy (Lap SG); (5) Laparoscopic conversions of previous SG, to RYGB. All of the operations mentioned above were performed between the years 1995 and 2011.

All patients included in the study satisfied the indications for metabolic and bariatric surgery as they were adapted over the years. The choice of operation was based on surgeon expertise for each operation, individual patient risk factors, patient preference, and specific goals (e.g., diabetes remission). In general, higher BMI and superobese patients were elected for malabsorptive procedures (RYGB and BPD). Within our institution, the preoperative diagnosis of diabetes was also a common indication for malabsorptive procedures, although a minority of patients were treated with SG. A BMI of more than 50 was also commonly used as a decisive factor for a patient to undergo a BPD procedure. Gastric fundus excision, as an addition to RYGB, was based on surgeon preference.

Prior to surgical consultation, each patient was thoroughly examined by a team physician, including cardiology and pulmonology specialists, in order to assess for comorbidities in addition to their assessment for anesthesia-related risks by the anesthesiologist. Nutritional specialists screened each patient for nutritional deficiencies and evaluated their dietary habits. The preoperative assessment was completed by the consultation of psychiatrists and clinical psychologists who confirmed the ability and willingness of each patient to make necessary behavioral adjustments. Finally, an independent committee of specialists confirmed adherence to indications for bariatric surgery from the NIH Consensus Development Panel [4], and approved each case for surgical intervention on a personalized basis.

Our surgical approach for each type of operation was guided by well-established international standards. The study cohort was divided into five groups depending on the type of operation. (1) For the laparoscopic Roux-en-Y Gastric Bypass (RYGB), we utilized between 110 cm and 150 cm of the jejunal small bowel as the biliopancreatic limb and 75 to 150 cm as the Roux limb. (2) For the Long-Limb modification of a biliopancreatic diversion (BPD-LL), we opted for alimentary limbs of more than 150 cm in addition to shorter common limbs (approx. 40–50 cm) in select patients that, according to the team’s evaluation, could benefit from more drastic weight loss, while not necessarily having a BMI > 50 kg/m^2^. (3) For the laparoscopic approach of RYGB with the fundus excision modification, we included gastric fundus excision, guided by the effect fundus resection can have on the reduction in leptin and ghrelin levels that is expected to hormonally facilitate weight loss. The election of gastric fundus excision was based on higher preoperative BMI levels. (4) For our Sleeve Gastrectomy (SG) technique, gastric dissection began 3 to 6 cm from the pylorus, according to current practice consensuses. The calibration of the gastric tube size was carried out using a 33 Fr bougie, following standard practice within our institution. (5) SG to RYGB conversions were performed on the basis of subpar weight loss following sleeve gastrectomy, or persistent SG-related adverse effects, the main one of which being GERD symptomatology.

All patients within our institution received routine postoperative nutritional supplementation of Thiamin (100 mg daily), Cobalamin (800 μg daily), Folate Acid (800–1000 μg daily), Vitamin D (1500–2000 mg/d depending on the operation), and Vitamins A (10,000 IU daily), E (15 mg daily), and K (120 μg daily). Iron supplementation was administered at a dosage of 45 mg to 60 mg of elemental iron daily for all bariatric patients as well. Iron supplementation was routinely administered among other micronutrients in the postoperative period, in accordance with the joint AACE/ASMBS/OMA/ASA guidelines on the perioperative nutrition support of bariatric surgery patients [4]. Unfortunately, patient adherence data were not available past the second postoperative year due to institutional factors.

The primary outcome of our study was set as the effectiveness of each surgical approach for weight loss. In order to quantify the level of effectiveness, we evaluated the following: (1) the average percentage of total weight loss for each of the included operations, and (2) the percentage of BMI loss following surgery.

In addition to our primary measures of weight loss after bariatric surgery, our follow-up included several secondary outcomes retrospectively recorded as an audit on bariatric surgery: (1) Long-term bariatric surgery-related complications: (a) nausea and (b) vomiting. (2) Resolution of diabetes defined as the return to normal HbA1C (below 6.5%) levels, without the need for medication for a minimum of three months. (3) Nutritional deficiencies. Protein malnutrition was also assessed but was not reported due to the small number of patients, which did not allow for any conclusions to be drawn.

In order to quantify and compare the severity and rates of adverse effects between the five different study populations, we utilized a four- or five-point Likert scale. This was decided upon by the research team members in order to minimize “neutral” responses as much as possible so that even minimal differences between the study populations could be assessed. The preoperative diagnosis of diabetes was defined as abnormal fasting glucose (above 100 mg/dL fasting glucose) or HbA1C (below 6.5%) levels. The comparison of continuous numerical outcomes (e.g., BMI, % of weight loss) was carried out using the ANOVA test for the comparison of means, due to the skewed nature of the data, or the Mann-Whitney U non-parametric test, for the comparison of two populations. Categorical outcomes (such as the percentage of patients that reported an adverse effect at a particular severity score) were analyzed using the chi-squared test for the comparison of proportions.

In order to determine the effect that each type of operation had on the outcomes, we utilized linear and logistic regression models to quantify the correlation between outcome and exposure. For outcomes that displayed statistical significance on univariate analysis, a second multinomial regression model was built in order to account for the effect that age, sex, preoperative weight, and preoperative BMI had on the observed differences. Despite the selected patients being accurately followed up, we encountered several instances of incomplete datasets, possibly due to changes in logging systems over the years. Missing data analysis did not reveal any patterns, and therefore, all available patients were included in the final analysis of each outcome. Statistical analysis was carried out using IBM SPSS Statistics 26 (IBM Corp., Armonk, NY, USA) and Jamovi (Jamovi project Version 1.6).

## 3. Results

Our retrospective data search returned a total of 1220 patients that underwent bariatric surgery in our department. Of them, 786 patients underwent RYGB, 162 underwent long-limb BPD, 89 underwent Laparoscopic RYGB with fundus excision, 116 underwent Laparoscopic SG, and 67 underwent Laparoscopic RYGB as a revision operation. Overall, 490 patient records had complete 10-year follow-up data, and were included in our retrospective study (40.1% overall follow-up rate). Of them, 145 (37.1%) were male patients and 345 (62.9%) were female. Surgical operations were carried out from 1995 until 2011, when the last set of patients with full 10-year postoperative data was available. Of these patients, 322 underwent RYGB (40.9% 10-year follow-up rate), 58 underwent long-limb BPD (35.8% 10-year follow-up rate), 34 underwent Laparoscopic RYGB with fundus excision (38.2% 10-year follow-up rate), 47 underwent Laparoscopic SG (40.5% 10-year follow-up rate), and 29 patients underwent Laparoscopic RYGB as a revision operation of prior SG (43.2% 10-year follow-up rate).

Among the latter population of patients, eight patients (28%) underwent conversion due to refractory GERD symptomatology, fifteen patients (51.7%) due to insufficient weight loss, and six patients for persistently reported dysphagia and gastric stenosis. The baseline characteristics of our study population and the total number of patients operated on can be seen in Table 1. No statistically significant differences were found in the reported baseline measures between the studied populations, or between baseline measures in the sum of operated patients, and patients included in the follow-up (*p* > 0.05).

### 3.1. Percentile Weight Loss

The average percentage of total weight loss (%TWL) was assessed for each operation. The mean percentage of postoperative weight loss after 10 years and the standard deviation for each type of operation can be found in Table 2. RYGB and Long-Limb BPD were able to produce a TWL of 37.6% (SD 10.6%) and 37.5% (SD 11.1) on average after 10 years of follow-up. Laparoscopic SG reduced the TWL of patients by 33.7% (SD 9.75), followed by the RYGB as a revision procedure after SG (32.1% SD 10.9). The least effective procedure for weight reduction was the laparoscopic RYGB with gastric fundus excision, measuring an average of 27.4% (SD 13.6) TWL. The aforementioned differences were all found to be statistically significant (*p* < 0.001).

We used multivariate regression to include the effect of age, gender, preoperative weight, and preoperative excess weight on the percentage of weight loss. The RYGB group of patients was used as the reference level (Appendix A). Autocorrelation, linearity, and normality assumptions were confirmed to be satisfied by the data. The F-test of overall significance confirmed a good model fit of the data.

Results from the multivariate analysis indicate that only the laparoscopic RYGB with gastric fundus extraction was an independent factor that negatively influenced weight loss outcomes when compared to other operations. In contrast, excess weight, age, and female gender were all found to be independently associated with better weight loss outcomes (Appendix A).

### 3.2. Percentage of BMI Loss at 10 Years of Follow-Up

Within our patient cohort, no statistically significant difference in the percentage of BMI loss of patients was found, between the different types of operations. Patients that underwent Laparoscopic SG were found to have the greatest percentile reduction BMI (37.7%, SD 11.4%), closely followed by patients that underwent RRGB, with an average reduction in BMI of 36.7% (SD 13.3%) and those that underwent Long-Limb BPD, with 36.5% (SD 13.7%). The observed differences were not statistically significant (*p* = 0.164), as seen in Table 2.

### 3.3. BMI after 10 Years Post Bariatric Surgery

The laparoscopic SG achieved the biggest loss in BMI units, at an average of 26 kg/m^2^ (SD 14.7), followed by the patients within the Long-Limb BPD group, which achieved an average loss of 25.2 kg/m^2^ (SD 10.5). After utilizing multinomial linear regression models, the excess weight, and preoperative BMI were confirmed as independent predictors for BMI loss after 10 years of surgery. The Long-Limb BPD and Laparoscopic SG were also independently associated with BMI loss (Appendix A).

### 3.4. Percentile Excess Weight Loss

The results for mean percentile excess weight loss (%EWL), can be found in Table 2. %EWL at 10 years of follow-up was higher for the long-limb variation of the BPD procedure with 82.8% (SD 42.8), followed by the RYGB procedure with 76.8% (SD 45.1) %EWL. The rest of the studied procedures produced inferior 10-year EWL outcomes. The Kruskal-Wallis ANOVA test revealed overall significance for the percentile differences with a *p* value of 0.001. The pairwise comparisons (Dwass-Steel-Critchlow-Flinger method), however, revealed that there were only two pairs of operation types that produced a statistically significant % of EWL at 10 years. The RYGB was found to be statistically superior to the gastric fundus variant, and the long-limb BPD was also superior to the same operation (Appendix A).

In the multivariate regression model, the % EWL was evaluated as a function of the type of operation, while adjusting for age, sex, preoperative BMI, excess weight (kg), and preoperative diagnosis of diabetes (Appendix A). The type of operation was the only factor independently associated with a higher % EWL at 10 years, meaning that the trend of significance in the univariate comparison of means holds true after adjusting for confounders of weight loss.

### 3.5. Long-Term Bariatric Surgery Adverse Effects on Feeding Tolerance

Self-reported nausea was significantly associated with the type of bariatric operation that patients underwent according to the chi-squared test analysis (*p* < 0.001). The majority of the patients reported little to no postoperative feelings of nausea at 10 year. RYGB and long-limb BPD patients reported no nausea in 66.5% and 84.5% of the instances, respectively, indicating a statistically significant association between low nausea percentages and the two types of operations.

Vomiting also differed between patients of different bariatric surgical approaches. Long-limb BPD patients reported no vomiting at 10 years after surgery in 84.5% of the patients. In addition, patients within the RYGB as a redo of the previous SG group reported no vomiting in 72.4% of the cases (Table 3), a difference with a *p*-value of 0.002.

### 3.6. Effects on Diabetes

Within our patient cohort, there were no statistically significant differences in the percentile of persistent diagnosis of diabetes at 10 years after surgery (*p* = 0.096). There were very few numbers of preoperatively diabetic patients to draw clinically significant conclusions. When our results on postoperative diabetes status were adjusted for age, preoperative diabetic status, preoperative weight, preoperative BMI, and type of operation, age was the only variable that independently influenced the postoperative diabetes status for our patients (Supplemental Appendix A).

### 3.7. Nutritional Deficiencies

No statistically significant differences were observed within our patient cohort regarding long-term deficiencies of several nutrients, commonly associated with bariatric operations. Serum iron levels were postoperatively evaluated at 10 years and showed no difference between our studied surgical approaches regarding acquired iron deficiency, as seen in Table 4 (*p* = 0.095).

Ferritin deficiency was also not found to be significantly associated with any one of the bariatric surgery approaches studied at our institution (*p* = 0.647). In addition, folic acid levels did not significantly differ between study cohorts (Table 4).

### 3.8. Laparoscopic Sleeve Gastrectomy Revisions

Within the retrospective cohort of our patients, we included 29 patients in total who completed 10 years of postoperative follow-up after being revised from LSG to RYGB. Indications for conversion included ineffective weight loss for the majority (51.7%) of the patients, refractory GERD in 28% of the patients, and lastly, postoperative anastomotic stenosis in 20% of the patients.

## 4. Discussion

Despite previous discussions on methods of weight loss, it is now widely accepted that the surgical approach results in superior weight loss outcomes compared to non-invasive weight loss measures [5,6]. An ongoing area of debate is whether there is an underlying difference between the various bariatric procedures regarding several postoperative outcomes, mainly weight loss. The comparison of patients that underwent RYGB, long-limb BPD, RYGB with gastric fundus excision, and sleeve gastrectomy reveals certain differences after 10 years of follow-up. Traditionally, biliopancreatic diversion procedures have been viewed as those with the most optimal weight loss outcomes [7,8].

Within our cohort of patients, long-limb BPD and RYGB seem to affirm their status as the optimal approach for weight reduction, regarding long-term outcomes. Previous studies on long-term outcomes of weight reduction surgery also confirm our observations regarding the difference in weight loss between diversion procedures and sleeve gastrectomy [9,10,11]. Additionally, other authors have noted that despite an observed superiority of SG procedures for medium-term weight loss, RYGB was superior in terms of both the short- and long-term aspects of weight loss [11]. This is also noted in our study, with RYGB being able to produce a significantly higher percentile of TWL as well as of EWL after 10 years. A recent meta-analysis of weight loss outcomes as a long-term result revealed no major differences between BPD and SG operations [12]. After the multivariate analysis, however, which assessed the effect that choice of operation had on weight loss, it was evident that there is a weaker association between long-term weight loss and the type of operation. Preoperative excess weight, younger age, and female sex were all found to be independent positive predictors of weight loss. None of the procedures studied in our cohort were independently associated with better crude weight loss outcomes on multivariable analyses.

On the contrary, when examining the percentile of excess weight loss, the type of operation was the only independent prognostic factor associated with %EWL. In our cohort, we were able to demonstrate a statistically significant superiority of BPD procedures regarding %EWL, even when compared to RYGB. Multivariable logistic regression also confirmed that the statistically significant differences in %EWL between BPD and RYGB were independently correlated with the type of procedure. However, there were very limited pairwise comparisons that resulted in significant differences in %EWL, although multivariate regression analysis indicated that operation choice was significantly associated in a positive manner with EWL. Pairwise comparison can also be influenced by the difference in patient numbers for each operation (e.g., between RYGB and BPD). Our results also show a disproportionate difference between %EWL and %BMI metrics as well. According to previously published studies, this is a frequently reported phenomenon and the reason behind the election of %EWL as the primary and most trustworthy weight loss measure for bariatric patients [13,14,15]. This was also observed within our analysis regarding %TWL, for which differences between BPD and RYGB were also small. As previously published, %TWL is significantly influenced by preoperative BMI, in contrast to %EWL [16], an effect that can also explain the independent association between %TWL and excess weight in the regression model (see “Percentile Weight Loss” in the Results section).

RYGB with gastric fundus excision was the only type of surgery that was found to have a significantly negative influence on weight loss compared to other procedures. Some bariatric surgeons have proposed an additional resection of the gastric fundus as a weight loss enhancer. The physiological principle behind this is stated to be the decrease in ghrelin, a component of the anorexigenic gut-brain axis hormones, through the removal of its main site of excretion [17,18]. Recent publications, however, seem to contradict this concept, since both gastric fundus removal, as well as lower ghrelin levels, do not seem to produce significant effects on weight loss outcomes [19,20]. Previous authors also found that RYGB and RYGB with additional fundus resection had similar weight loss outcomes [21]. The study by Delko et al. [21] found that fundus resection not only did not contribute to weight loss but also did not alter patient-reported appetite and satiety. Ghrelin is peripherally produced in the gastric antrum and pancreas, and centrally in the pituitary gland, all of which could be compensating for the loss of secretagogue function of the fundus. Another mechanism that could explain the similarity between RYGB and RYGB with fundus excision could be that the weight loss is largely attributed to the common malabsorptive effects of the two procedures, much more than to proposed hormonal changes. Additionally, a study on unintentional fundus retention commenting on weight loss after SG also found no difference in patient outcomes [21]. Clinical studies that challenge the role of ghrelin in weight loss are becoming more abundant, with some reporting a negative effect of ghrelin levels in weight loss after gastric bypass [22,23,24]. To the best of our knowledge, ours is the first study reporting 10-year weight loss outcomes on bypass patients with fundus resection. Long-term changes in ghrelin levels and weight loss after fundus resection prove that not only do ghrelin levels postoperatively normalize after several months [25], but lower ghrelin levels do not correlate with better weight loss outcomes [25]. Ghrelin and leptin secretion are closely correlated, as is evident by the similar expression of neuronal receptors in the hypothalamic regions [24]. Induced hypo-leptinemia has been shown to negatively influence ghrelin secretion and induce ghrelin resistance [26]. It can be hypothesized that hypo-ghrelinemia associated with gastric fundus resection negatively influences leptin secretion, leading to greater caloric consumption, and worse weight loss outcomes or weight regain.

Our analysis, despite producing statistically significant differences regarding the %TWL and %BMIL between different types of bariatric surgery, also needs to be viewed from a clinical perspective. With the exception of the noted difference regarding the laparoscopic RYGB with gastric fundus excision, there are only small percentile differences in weight loss after 10 years of bariatric surgery. These results support the current approach of comparable weight loss outcomes between most of the approved weight loss procedures that are commonly reported in the literature [8,27,28]. When examining the units of BMI reduction after 10 years of follow-up, we can see that they are largely in line with our findings of the percentile of weight loss. After the multivariate analysis, however, only preoperative BMI levels, long-limb BPD, and SG were independently associated with a higher BMI difference after 10 years. In addition, a more sensitive indicator of weight loss, the %BMI loss, rather than crude BMI loss, was not found to significantly differ between different operations. Expanding on the % of BMI loss, our results confirm those of previous comparative studies and meta-analyses, that ultimately did not showcase significant differences after 5 years of follow-up between LSG and RYGB [9,28,29,30,31]. Higher BMI levels have previously been associated with more favorable weight loss outcomes, an observation that is confirmed by our findings regarding long-term weight loss outcomes [17,32,33]. Notably, our study is one of the few that reports a higher BMI to be independently associated with better weight loss outcomes after such a long-term follow-up period, since authors have usually generated results after 12, 24, or 36 months of follow-up [16].

Patient selection in bariatric surgery is a crucial component of the discussion regarding weight loss outcomes. Over the years, indications for selecting appropriate bariatric surgery candidates have been evolving, with the latest edition of the ASMBS/IFSO guidelines [4] recommending bariatric surgery for individuals with a BMI of 35 or above, or a BMI of 30–34.9 with a history of metabolic disease, and includes alterations for select patient populations. Oftentimes, however, patient selection for bariatric surgery is more complicated, even within the confines of said guidelines, and can therefore impact weight loss outcomes. For instance, a survey among 820 bariatric surgeons showed that they were less likely to offer bariatric surgery to older patients (OR 0.70), those with poor functional status as a result of medical comorbidities (OR 0.66), poor social support (OR 0.37), and factors regarding reimbursement for the operation [34]. Bariatric surgery candidates are extensively screened for additional predictors of success in weight loss, including the ability to adapt their diet to the postoperative requirements, pre-existing psychiatric conditions, and success in previous attempts to maintain beneficial weight loss for a period of time [35]. These screening criteria, although useful in appropriately selecting patients by balancing the risks of the operation with superior weight loss outcomes and comorbidity resolution, can be seen as excluding certain patient populations and predetermining optimal candidates for higher percentages of weight loss [36,37]. Additionally, surgeons have reported a history of malignancy, high risk of thromboembolic events, a BMI > 70, and poor functional status as relative contraindications for their practice, although not mentioned as absolute contraindications in current recommendations [38]. On the contrary, when discussing non-surgical management strategies for obesity, these are offered to a wider pool of patients [37], including those with unfavorable characteristics for effective weight loss. This constitutes a source of bias, not specific to the present study, but for weight loss outcomes following bariatric surgery as a whole.

In one meta-analysis that compared diabetes resolution between different procedures, BPD achieved remission of diabetes in 95.1% of the patients, compared to 80.3% in RYGB and 79.7% in SG [39]. Mingrone et al. produced similar results, with BPD achieving glycemic control [40]. The Swedish Obese Subjects study concluded that surgical approaches are superior in glycemic control, when compared to oral treatments, after 15 years of follow-up [41]. It did, however, lack in providing a comparison between the different operative strategies. Our data indicate that, after 10 years, the effect of specific bariatric operations might be diminished. One possible explanation for this observation is that after such long follow-up periods, patients have fully adjusted to a different eating lifestyle, or that, given enough time, the bariatric operations studied here manage to produce similar effects on diabetes. Multivariate analysis also failed to detect any patterns, regarding glycemic control and the type of operation. Our cohort suffered from a lack of sufficient patient numbers to draw clinically significant conclusions. Nevertheless, we were able to affirm previous results that showcase high-resolution rates for malabsorptive operations (RYGB and BPD).

Iron, ferritin, and folic acid deficiency studies failed to indicate any intra-operational difference after 10 years of follow-up. Since iron is primarily absorbed in the duodenum and jejunum, bypass procedures are commonly associated with iron deficiency [5,42,43]. Ferritin deficiency patterns closely follow those of iron deficiency. Previous reports comparing ferritin and iron deficiency seem to conclude that they are more prevalent in GBRY and BPD procedures [44,45]. Once again, however, the current literature lacks data for long-term follow-up of nutritional deficiencies, making our results an indication that known modifications of dietary micronutrients need to be compared between surgical approaches in long-term follow-up periods. One important point to be raised is the role of obesity as a factor of systemic inflammation that is capable of altering iron metabolism, as previously stated [46]. The resolution of obesity after 10 years might be one of the reasons our cohort failed to produce inter-operational differences. Folate absorption mainly relies on the functionality of the remaining absorptive small bowel surface, the adaptation of which could explain the lack of observed differences. In general, there is a less clear picture in the literature regarding folate deficiency in different bariatric procedures, since even in malabsorptive procedures the remaining small bowel achieves adequate absorption [47], with authors reporting more significant deficiency for all types of operations [46]. Another important factor to consider is the nutritional supplementation of postoperative bariatric patients, which was employed as routine practice in our institution, and could be the reason behind the lack of statistically significant differences. However, it must be noted, that due to a lack of routine adherence review past the second postoperative year, we were not able to procure data on patient adherence that would influence deficiencies.

### Study Limitations

Our retrospective study has several limitations. Firstly, the lack of randomization creates a possible source of bias in our patient allocation. In addition, while operations were carried out under one institution, they were done so over a long time period, meaning individual surgeons and surgical strategy evolved over time, influencing the choice of operation on individual patients. Additionally, the utilization of self-reporting measures as part of the follow-up procedure also leaves room for recall bias on behalf of the patients. Compliance with nutritional supplements was not assessed. This would affect the nutritional outcomes and the association between nutritional deficiencies and behavioral characteristics of our cohort. The variation in individual nutritional limb lengths was also not able to be assessed, due to a lack of reporting over the years. The present study was also focused on reporting adverse outcomes of bariatric surgery regarding oral feeding tolerance. Other, more serious adverse events such as internal hernias and ulceration are beyond the scope of the study. Regarding nutritional deficiencies, there were no available data on patient adherence to supplementation past the second postoperative year. Lastly, there is an apparent underreporting of sleeve gastrectomy patients when compared to other similar studies, which makes the reporting of significant outcomes more difficult. This can be attributed to both institution-wide preference for bypass procedures, as well as a great percentage of loss of follow-up for such patients. Aside from these limitations, the presented study remains a large single-center bariatric study with 10 years follow-up, with a considerable number of patients, that is expected to add to the existing literature on the long-term follow-up of bariatric patients.

## 5. Conclusions

In this study, we presented data after 10 years of follow-up of bariatric patients, regarding a wide array of outcomes and adverse effects. There were significant differences in weight and BMI reduction between surgical approaches, as well as in reported appetite and adverse effects. Glycemic control and nutritional deficiencies did not significantly differ between the types of operation after 10 years of follow-up. Our study indicates that longer follow-up times might be the key to interpreting known short-term differences between different bariatric operations.

## Figures and Tables

**Table 1 jcm-12-04973-t001:** Baseline Patient Characteristics (including loss to follow-up).

	LAP RYGB (Follow-Up N = 322)	LAP RYGB (Sum of Patients)	BPD-LL (Follow-Up N = 58)	BPD-LL (Sum of Patients)	LAP-RYGB with Gastric Fundus Excision (Follow-Up N = 34)	LAP-RYGB with Gastric Fundus Excision (Sum of Patients)	LAP-Sleeve Gastrectomy (Follow-Up N = 47)	LAP-Sleeve Gastrectomy (Sum of Patients)	LAP SG to RYGB Conversion (Follow-Up) N = 29	LAP SG to RYGB Conversion (Sum of Patients)
Gender (n)										
Male	94	259	18	56	6	22	17	38	10	25
Female	228	527	40	106	28	67	30	78	19	42
Age (years)	48 ± 10.1	44.2 ± 12.4	47.9 ± 10.7	51.2 ± 13.4	46.2 ± 11.1	48.7 ± 16.2	47.4 ± 11.1	44.2 ± 13.4	51.3 ± 8.64	49.2 ± 9.7
Preoperative Weight (kg)	142 ± 30.8	149 ± 37.1	138 ± 28.9	129 ± 26.7	142 ± 34.9	146 ± 37.2	139 ± 27.2	145 ± 35.4	138 ± 26.4	137 ± 28.2
Preoperative BMI (kg/m^2^)	52.7 ± 9.03	58.2 ± 10.9	51.9 ± 8.04	52.5 ± 9.2	49.3 ± 7.09	52.8 ± 9.2	51.1 ± 8.3	49.2 ± 9.4	50.8 ± 8.73	48.3 ± 10.2
Excess Weight (kg)	82.7 ± 26	80.4 ± 21	81.6 ± 23.4	84.5 ± 26.5	85.2 ± 30.7	91.6 ± 37.3	80.3 ± 20.1	84.2 ± 25.3	82.1 ± 24.3	86.2 ± 26.8
Diagnosis of Diabetes (%)	29 (9.0%)	68 (8.6%)	7 (12.1%)	25 (15.4%)	2 (5.9%)	7 (7.8%)	4 (8.5%)	13 (11.2%)	3 (10.3%)	8 (11.9%)

**Table 2 jcm-12-04973-t002:** Average %BMI loss and %EWL after bariatric surgery at 10 years of follow-up.

Type of Operation	Number of Operations	Mean %TWL	*p*-Value *	Mean %BMI Loss	*p*-Value *
RYGB	322	37.6	<0.01	36.7	0.164
BPD-LL	58	37.5		36.5	
Lap RYGB with fundus excision	34	26.4		30.8	
Lap SG	47	33.7		37.7	
Lap SG to RYGB conversion	29	32.1		34.7	

* Referring to the corresponding Kruskal-Wallis test for statistical significance of average weight loss metrics.

**Table 3 jcm-12-04973-t003:** Outcomes on nausea and vomiting at 10 years post bariatric surgery.

Nausea						
Nausea Grading	BPD with GBRY	LAP-RYGB-LL with Gastric Fundus Extraction	BPD-LL	LAP-Sleeve Gastrectomy	LAP SG-Redo-LAP RYGB	Total
0	214 (66.5)	16 (47.1)	49 (84.5)	21 (44.7)	21 (72.4)	321 (65.5)
1	97 (30.1)	13 (38.2)	9 (15.5)	23 (48.9)	7 (24.1)	149 (30.4)
2	10 (3.1)	2 (5.9)	0 (0.0)	3 (6.4)	0 (0.0)	15 (3.1)
3	1 (0.3)	3 (8.8)	0 (0.0)	0 (0.0)	1 (3.4)	5 (1.0)
**Total**	322	34	58	47	29	490
**Vomiting**						
**Vomiting Grading**	**BPD with GBRY**	**LAP-RYGB-LL with Gastric Fundus Extraction**	**BPD-LL**	**LAP-Sleeve Gastrectomy**	**LAP SG-Redo-LAP RYGB**	**Total**
0	209 (64.9)	13 (38.2)	49 (84.5)	21 (44.7)	21 (72.4)	313 (63.9)
1	98 (30.4)	14 (41.2)	9 (15.5)	24 (51.1)	7 (24.1)	152 (31.0)
2	14 (4.3)	3 (8.8)	0 (0.0)	2 (4.3)	0 (0.0)	19 (3.9)
3	1 (0.3)	4 (11.8)	0 (0.0)	0 (0.0)	1 (3.4)	6 (1.2)
**Total**	322 (100)	34 (100)	58 (100)	47 (100)	29 (100)	490 (100)

**Table 4 jcm-12-04973-t004:** Iron, ferritin and folic acid deficiencies at 10 years post bariatric surgery.

	RYGB	LAP-RYGB-LL with Gastric Fundus Excision	BPD-LL	LAP-Sleeve Gastrectomy	LAP SG-Redo-LAP RYGB	*p*-Value *
Iron Deficiency	133	10	15	14	3	0.095
No Deficiency	161	21	27	19	15	
Ferritin Deficiency	150	13	20	20	9	0.647
No Deficiency	132	14	25	13	10	
Folic Acid Deficiency	150	13	20	20	9	0.254
No Deficiency	132	14	25	13	10	

* Referring to the corresponding chi-squared test for statistical significance.

## Data Availability

Study data can be provided after requested from the authors.

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
