# Peer review of "Outcomes at 10-Year Follow-Up after Roux-en-Y Gastric Bypass, Biliopancreatic Diversion, and Sleeve Gastrectomy"

_jcm, 2023, doi:10.3390/jcm12154973_

Round 1

Reviewer 1 Report

Thanks for the opportunity to read this interesting manuscript.

This is a retrospective study comparing long-term outcomes of 5 baritric procedures in a single center.

The subject of the study is interesting and assess an important issue in the surgical treatment for morbid obesity.

Major issue:

1. Currently the two major options to report weight loss post Bariatric Surgery are %TWL and %EBMIL . The option of %EWL are no longer reported in major bariatric surgery journal. I would focus on your results on the 2 popular option.

2. The definition of remission of diabetes is not clarify . There're a few options to define diabetes remission and you need to clarify your method.

 Some minor comments are listed below.

Line 44 - English need to be revised in the sentence 

Line 77 - Reference is needed. 

Line 88 - Indication for BPD need to be clarified

116 - define normal level of Glucose and HbA1C

Table 1 - Does the groups study comparable - Statistical analysis is needed

line 237 - Please clarify the sentence

Nutritional Deficiencies - Pleaase clarify post operative vitamin treatment. Did you evaluate the vitamin compliance at 10 years ? Need to be written in the result section as well.

was written in the comments section

Author Response

Dear Reviewer,

Thank you kindly for taking the time to review our manuscript and consider it for publication. It is our honor to receive positive commentary from field experts, including insightful comments to better our work.

Your suggestions have been incorporated in their entirety within our manuscript and we firmly believe that they have augmented the quality of our paper.

In detail regarding the two major issues:

  • Thank you for underscoring this difference between current trends in reporting metrics. We appreciate your commentary and focused more on TWL and EBMIL. Table 2 now includes only these metrics, which have also been prioritized in the Results section, with only brief mentions of EWL results. In addition, we significantly shortened the corresponding section in the Discussion, regarding EWL.
  • We have now clearly defined both the preoperative diagnosis of diabetes, and the remission criteria (according to the most recent consensus statement from the American Diabetes Association, the Endocrine Society, the European Association for the Study of Diabetes and Diabetes UK) in Lines 119-130.

Regarding the minor issues:

  • References and language revisions are included as pointed out.
  • We have better defined the operation selection criteria in Lines 69-77
  • Statistical comparison of all groups has been added in Table 1, which was expanded to fit the requests of both reviewers regarding the sum of the patients. Statistical measures can be seen in the Results section, regarding baseline characteristics, as they would not fit in the Table without further cluttering.
  • Postoperative vitamin administration is now better described in Lines 105-114. Unfortunately complete data on adherence after 10 years were not available. this has been added to the corresponding Discussion section, as well as in the study limitations.

Once again, thank you for your consideration.

Kind regards,

The authors

Reviewer 2 Report

Thank you for the opportunity to review the manuscript: “Outcomes at 10 Year Follow-up after Roux-en Y Gastric Bypass, Biliopancreatic Diversion and Sleeve Gastrectomy”. In this retrospective cohort 490 patient underwent bariatric surgery and had 10 years follow-up. I think there are some flaws in the study design and some other issues in the manuscript:

1. The manuscript should contain ethical statement. Was this study approved by a medical ethical committee or local board?

2. The statistical analysis is not sound and some of the groups are just too small. In some of the tables no p-values are mentioned.

3. I cannot see what this study adds. The authors state that this is one of the largest long-term follow-up studies. In my opinion this sounds perky. In fact, there are many larger series with 10 year follow-up.  

4. If results after RYGB and BPD are superior, I think this is mainly due to patient selection. The specific selection criteria for a certain type of operation are not mentioned in the manuscript. I think those criteria would be the most interesting to learn and it would be beneficial to mention them.

5. The supplemental material contains a comment?

Needs minor spell check. 

Author Response

Dear Reviewer,

Thank you kindly for taking the time to review our manuscript and consider it for publication. It is our honor to receive positive commentary from field experts, including insightful comments to better our work.

Your suggestions have been incorporated in their entirety within our manuscript and we firmly believe that they have augmented the quality of our paper.

In detail regarding the major issues:

  • The approval number by the Bioethics committee is now added in the first paragraph of the materials and methods section.
  • Indeed some of the included patient subgroups were too small for clinically significant results, even when producing statistically significant results. The utilization of each statistical test is now more clearly stated in the text at each appropriate section and p-values have been added in all tables.
  • Thank you for your observation. Indeed, this may have been a slight exaggeration on our part, only intended to showcase that this is one of the relatively few studies that looks into a high volume of patients for a long-term follow-up. We have appropriately amended our statement while acknowledging existing literature.
  • Patient selection criteria are now added in Lines 69-77. This is a keen observation that baseline patient characteristics can have an impact on weight loss. In order to augment out arguments, we have now updated Table 1 with the baseline characteristics of all patients operated in our department, including those lost to follow-up. Statistical analysis (1st paragraph of Results) indicated no difference in baseline characteristics either between patient subgroups, or between those included in the analysis and lost to follow-up. In addition, within the multivariable models regarding weight loss, all major patient confounders are included and adjusted for, and on certain occasions found to be independent predictors for weight loss.
  • Apologies for this mishap, it is now amended.

Once again, thank you for your consideration.

Kind regards,

The authors

Round 2

Reviewer 2 Report

Thank you for the opportunity to review the revised version of the manuscript. The authors made several corrections to improve the overall quality manuscript. All questions and suggestions were noted by the authors. Except one. 

I think good overall outcome has to do with patient selection. The authors state that this is discussed in the article, because there are hardly any baseline differences between groups. 

What I was suggesting is the outcome is less dependent on the surgical technique. It depends on who is selected for bariatric surgery and who is not. This needs to be discussed. 

The overall English is improved and needs some minor spell check. 

Author Response

Dear Reviewer,

Thank you kindly for once again taking the time out of your schedule to review our manuscript and assess the changes made. We would like to thank you for your positive comments on our work and inform you of the latest changes to the manuscript, as per your suggestion.

Indeed when discussing weight loss outcomes for bariatric patients, one more often than not pays little attention to the patient selection criteria, not just between operations, but for offering bariatric surgery as a whole (one could call this a form of "external validity"). We have realized the exact nature of your previous query, which we felt was the stimulus to expand our discussion on this issue.

We have discussed the issue of the effect of patient selection on weight loss outcomes, especially when opposed to nonoperative management which is offered to the vast majority of the obese population with far fewer restrictions. This part, along with references from the most recent studies addressing this issue, is added to the Discussion section, in Lines 365 to 390.  

We would like to thank you once again for aiding us in bettering our work, as there are very few papers discussing this inherent form of selection bias that should be considered when reporting weight loss outcomes.

Kind regards,

The authors
